# Divergent functions of late ESCRT components in *Giardia lamblia*: Insights from subcellular distributions and protein interactions

Nabanita Patra[1☯], Nabanita Saha[1☯¤], Trisha Ghosh[1], Babai Hazra[1], Shirsha Samanta[1], Pritha Mandal[1], Avishikta Chatterjee[1], Abhrajyoti Ghosh[1], Sandipan Ganguly[2], Srimonti Sarkar[1]*

1 Department of Biological Sciences, Bose Institute, Sector V, Bidhannagar, West Bengal, India,
2 Division of Parasitology, National Institute for Research in Bacterial Infections, Beliaghata, Kolkata, West Bengal, India

☯ These authors contributed equally to this work.
¤ Current Address: Michigan Medicine, University of Michigan, Ann Arbor
* srimonti@jcbose.ac.in

## Abstract

*Giardia lamblia*, a human gut pathogen, possesses a minimal ESCRT (Endosomal Sorting Complex Required for Transport) machinery. Paradoxically, there are multiple paralogues of some late-ESCRT components- three paralogues for Vps4, GlVps4a, GlVps4b, and GlVps4c, and two for Vps46, GlVps46a and GlVps46b. This study addressed whether these paralogues can potentially discharge distinct cellular functions by determining the subcellular distribution of the paralogues in trophozoites and during encystation. Consistent with the distribution of orthologues from model organisms, most of these components were found to be associated with various cellular membranes, particularly in regions of acute membrane bending. Some of these paralogues are also associated with microtubule structures, such as cytoplasmic axonemes and the median body. Considering their diverse sub-cellular distributions, it is likely that they perform non-overlapping functions within the cell. Further, their redistribution during encystation indicates that they may play a role in the morphological and functional changes accompanying this transition. The study also characterized GlIst1, an ESCRT-III accessory protein that undergoes unique post-translational myristoylation at lysine 43. GlIst1 selectively interacts with GlVps4b through non-canonical MIT-MIM interactions. GlIst1 also exhibits selective interaction with GlVps46b. Such selective interaction of GlIst1 with only specific paralogues of GlVps4 and GlVps46 further underscores the distinct cellular roles of these late-ESCRT paralogues.

**Data availability statement:** All relevant data are within the paper and its supporting information files. We have provided all raw data in figshare repository. The DOI is provided below 10.6084/m9.figshare.30361702 giardia ESCRT, published by Srimonti Sarkar on 2025-10-15 "Understanding ESCRT machinery in Giardia lamblia".

**Funding:** This work was supported by Bose Institute (R/16/19/1620 to SS). SS received her salary from her employer, Bose Institute. Bose Institute is an autonomous research institute fully supported by the Department of Science and Technology, Govt. of India. The mandate of this institute is the promotion of basic research. Fellowship support: N.P from Council of Scientific & Industrial Research (CSIR) (09/015(0534)/2019-EMR-I)N.S from University Grants Commission (UGC) (F.2-8/2002 (SA-1)T.G from University Grants Commission (UGC) (721/(CSIR-UGC NET DEC.2018)B.H from University Grants Commission(UGC) (221610061747) S.S from University Grants Commission (UGC) (221610185679)P.M from Council of Scientific & Industrial Research (CSIR) (09/015(0545)/2019-EMR-I)A.C from Department of Science & Technology (DST)-INSPIRE (IF180748). The funders had no role in the study design, data collection and analysis, decision to publish, or preparation of the manuscript.

**Competing interests:** The authors have declared that no competing interests exist.

## Author summary

*Giardia lamblia*, a unicellular protozoan parasite, manifests in two morphologically distinct forms- trophozoites and cysts. Transformation between these forms is essential for the organism's survival both within the host and without. It requires extensive membrane remodeling, which is likely to involve the Endosomal Sorting Complex Required for Transport (ESCRT) complexes, as this machinery is known to participate in both prokaryotic and eukaryotic membrane remodeling events. *Giardia* was known to encode multiple paralogs of the late-ESCRT components, GlVps4 and GlVps46, raising the possibility of them performing nonredundant functions. Our study indicates functional divergence of the paralogs by showing their nonoverlapping cellular distribution and also documenting selective interactions between them. The redistribution of these paralogs to sites of membrane deformation during encystation, along with regions of microtubule enrichment, indicates that they are likely to contribute to the stage transition process. Also, observed *Giardia*-specific interactions within ESCRTs opens new therapeutic avenues.

## Introduction

Membrane deformation is an integral part of several cellular processes, and the ESCRT (Endosomal Sorting Complex Required for Transport) machinery is one of the prime architects of several membrane remodeling events [1]. It plays a pivotal role in diverse cellular processes, including multivesicular body (MVB) formation, tubular endosomal trafficking for the recycling of receptors, cytokinesis, viral budding, plasma membrane repair etc. [2,3]. It was first discovered in *Saccharomyces cerevisiae*, where a screen for vacuolar protein sorting mutants identified many of the constituents of this machinery [4]. Subsequent studies have shown that ESCRT-mediated membrane remodeling is evolutionarily conserved as ESCRT-related components are present in archaea and bacteria [5,6]. This indicates that the origin of the ESCRT machinery predates the emergence of eukaryotes.

The Membrane recruitment of ESCRT components is sequential, and the precise orchestration of this process ensures efficient cargo sorting and membrane remodeling [7]. In the case of endosomal sorting, while the early ESCRT complexes (ESCRT-0, -I and -II) are involved in the recognition and subsequent aggregation of cargo proteins, the ESCRT-III complex, which functions downstream of the first three, plays a pivotal role in the final membrane remodeling steps [8,9]. For this purpose, ESCRT-III components polymerize into spiral filaments on the membrane surface [10]. Sequential recruitment and removal of various ESCRT-III components cause a change in the shape of this spiral, which facilitates membrane invagination and scission [11,12]. This membrane deformation and subunit exchange is fueled by ATP hydrolysis by the AAA-ATPase Vps4 and its associated proteins that form the Vps4

complex [13]. This stepwise assembly and disassembly of ESCRT components on the membrane ensures the precise spatial and temporal control of membrane remodeling events [14,15].

It is important to note that not all ESCRT-dependent cellular processes require the participation of all the ESCRT components. For example, ubiquitinated cargo recognition on the endosomal surface and the subsequent sorting into the vacuole lumen requires ESCRT-0 and –I [16,17]. However, plasma membrane repair, cytokinetic abscission, virus and microvesicle budding from the plasma membrane occur without the involvement of ESCRT-0 and ESCRT-I; instead, they are dependent on the ESCRT-III and Vps4 complexes, which together are sufficient for constricting and severing the narrow membrane neck [18]. Furthermore, certain membrane deformation events can occur with some, but not all, of the ESCRT-III components. For example, Ist1 and CHMP1A/CHMP1B (Vps46) are the main ESCRT-III family members driving cytokinesis [19], while CHMP4 (Snf7) and CHMP2 (Vps2) filamentous structures drive membrane scission during HIV budding from mammalian cells and the repair of tears in the plasma membrane involves CHMP2A, CHMP3 (Vps24), CHMP4B along with Vps4 [20,21]. Membrane reshaping by ESCRT-III components can happen even without Vps4, as is the case of sorting of mannose 6-phosphate and transferrin receptor through the formation of endosomal tubules [2]. The formation of these endosomal tubules requires the late-ESCRT components, Ist1 and CHMP1B, which function in conjunction with spastin, to efficiently divide endosomal tubules [2]. Spastin is similar to Vps4 as it contains both a AAA ATPase and a Microtubule Interacting and Trafficking (MIT) domain; it uses the latter to interact with the aforementioned ESCRT components [22,23]. However, unlike Vps4, it has the ability to severe microtubules [24]. Thus, the ESCRT machinery driving membrane sculpting is highly modular, wherein only some components can function at a certain cellular location to bring about the desired outcome.

The complexity of the ESCRT machinery has evolved with increasing complexity of organisms. In general, complex life forms have ESCRT complexes that are composed of more components, including many that are paralogous, while simpler organisms have less elaborate ESCRT machinery having fewer components. One of the simplest eukaryotic ESCRT machinery is that of *Giardia lamblia* (also termed *G. intestinalis* or *G. duodenalis*), a human gut pathogen [25]. This parasite has two morphologically distinct forms: trophozoites and cysts [26,27]. The transition between them is crucial for its survival within the host and transmission from one host to another [28]. This transformation necessitates extensive membrane restructuring [29,30], and the ESCRTs are likely to be involved in such processes. Previous research indicates that *G. lamblia* possesses a minimal ESCRT machinery, lacking the early ESCRT complexes, ESCRT-0 and ESCRT-I [31]. In addition, the ESCRT-III complex of *G. lamblia* comprises fewer core and accessory components, including Vps20, Vps2, and Vps24 as core ESCRT-III components, and Ist1 and the paralogs of Vps46 as accessory components. ESCRT-III subunits Snf7, Vps60, and Vta1 are likely to be absent in this parasite. However, some late-ESCRT components are present in multiple copies, notably Vps4 (GlVps4a, GlVps4b, and GlVps4c) and Vps46 (GlVps46a and Vps46b) [25,31]. As mentioned previously, ESCRT-dependent membrane deformation can proceed without the entire set of ESCRT proteins. Hence, the absence of certain ESCRT components in *Giardia* is unlikely to render the ESCRT-dependent process non-functional in this parasite.

The presence of multiple paralogs of GlVps4 and GlVps46 suggests that these paralogs may discharge overlapping and/or distinct functions. As spatial segregation within cells are indicative of specialized roles, we have carried out immunolocalization of these paralogs in trophozoites and encysting trophozoites. The results showed that these proteins are localized to various membrane-deforming sites and microtubule-rich regions. They mostly occupy unique sites within the cell, except for a few regions, such as the peripheral vesicles (PVs), the ventral disc (VD), flanges and the flagellar axonemes, where more than one of these components are present. In addition, we report the characterization of the Ist1 ortholog in *Giardia* (GlIst1) and show that it undergoes myristoylation at an internal K residue that is unique to the *Giardia* protein. Our results also document that GlIst1 selectively interacts with only one Vps4 and one Vps46 paralog. This selective interaction of GlIst1 with GlVps4b and GlVps46b underscores the distinct functional roles of these paralogues.

## Methods

### Ethics statement

All antibodies used in this study were procured from a contract research organization, Bio Bharati Life Science (Kolkata, India), using their custom antibody service. The purchase orders corresponding to these procurements are BI/19–20/3104/Biochem, BI/19–20/3102/Biochem and BI/19–20/3103/Biochem.

### Axenic culture of *G. lamblia*

The axenic culture of *G. lamblia* (ATCC 50803/WB clone C6) trophozoites in TY-I-S-33 medium was carried out as previously described [32] and encystation was induced according to the protocol of [33].

### Construction of plasmids

For the yeast two-hybrid assay, the following ORFs were PCR amplified, using gene-specific primers listed in S1 Table, from *G. lamblia* genomic DNA: GL50803_101906 (*glvps4a*), GI50803_16795 (*glvps4b*), GI50803_15469 (*glvps4c*), GI50803_15472 (*glvps46a*), GI50803_24947 (*glvps46b*) and GI50803_0011129 (*glist1*). The PCR products were cloned into vectors pGAD424 and/or pGBT9 (Takara). Primers used for cloning various fragments of *glist1* and *glvps4b* in these vectors are also listed in S1 Table, with restriction sites underlined in each primer and stop codons introduced where required for yeast two-hybrid assays. Details of the primers used in the co-purification assay for *glist1*, $glvps4b_{1-191}$, and *glvps46b*, for cloning into pET24dHisTEV and pETduet-1, are mentioned in S1 Table. For co-purification studies, the coding regions for $GlVps4b_{1-191}$ and GlVps46b were initially PCR amplified from *Giardia* genomic DNA using the forward and reverse primers listed in S1 Table. For cloning, the pETDuet-1 vector was linearized with NcoI and KpnI, while the PCR products were digested with BspHI and KpnI, due to the presence of an internal NcoI site in both the inserts, to generate compatible cohesive ends. Following gel purification, the inserts and vector were ligated and transformed into *E. coli*. Verified constructs were subsequently transformed into *E. coli* BL21 for protein co-expression and co-purification studies. The yeast genes, *VPS4* and *IST1* were PCR amplified from *S. cerevisiae* genomic DNA using specific set of primer pairs (S1 Table) and cloned into pGAD424 and pGBT9, respectively. To generate antibodies for immunolocalization, *glvps4b* and *glvps4c*, and *glvps46b* were PCR amplified using genomic DNA of *G. lamblia* (primers listed in S1 Table) and cloned into the expression vector. The constructs used for live-cell imaging in yeast were created by PCR amplifying of *glist1*, yeast *IST1* and yeast *VPS27* from the respective genomic DNA with the corresponding primers (S1 Table) and cloned into pGOGFP426 or pRS425-RFP vectors. All clones were sequenced to confirm the presence of the inserts. Details of all constructs are given in S2 Table.

### Protein expression and generation of antibodies

His-tagged GlVps4b was expressed in *E. coli* by induction with 0.5 mM IPTG at 30°C for 4 h. Following induction, cell lysis was carried out by sonication (20 s pulses followed by 1 min gap at 75% amplitude). The extract was analyzed by SDS-PAGE to ensure the induction of the desired protein, and the His-tagged protein was purified by using Ni-NTA (Qiagen). Similarly, His-tagged versions $GlVps4c_{1-261}$ and GlVps46b were expressed in *E. coli* by induction with 0.5 mM IPTG for 16 h at 20°C. The 6x-His-tagged proteins were purified as mentioned previously and purified proteins were handed over to Bio Bharati Life Science for generation of the respective antibodies in rabbits.

### Protein extraction and western blotting

*G. lamblia* extract was prepared by resuspending in lysis buffer (50 mM Tris-Cl, 100 mM NaCl, 2% SDS, 1% Triton X-100, pH 8.0), followed by incubation on ice for 30 min. The protein fraction was then obtained by centrifuging at 12,000 rpm for 10 min, and the resulting supernatant was subjected to Bradford assay for quantification of extracted protein. For

western blotting a 1:5000 dilution was used for antibodies against GlVps4a, GlVps4b, GlVps4c and GlVps46b, whereas a 1:8000 dilution was used for the antibody against GlVps46a. All dilutions were done in 1X TBS containing 0.2% BSA. After incubating the membranes with the respective primary antibodies for 2 h, they were washed three times with 1X TBST and twice with 1X TBS. Following the washing, alkaline phosphatase (AP)-conjugated anti-rabbit antibody was used at a dilution of 1:5000 for GlVps4a, GlVps4b, and GlVps4c, while anti-rabbit antibody conjugated to AP was used at a dilution of 1:10000 for GlVps46a and at a dilution of 1:8000 for GlVps46b. After incubating with the secondary antibodies for 2 h, the membranes were washed three times with 1X TBST for 5 min each, followed by two washes with 1X TBS. The blot was developed using NBT/BCIP (Thermo Scientific). Similarly, total protein from yeast harbouring BD-GlVps46a, BD-GlVps46b was prepared by resuspending the transformants in suspension buffer (1 M Tris-Cl pH 7.5, 0.5 M EDTA, 2.5 M NaCl, NP-40, 1 M DTT, of 0.1 M PMSF and protease inhibitor cocktail) and vortexing in the presence of glass beads for 10 min at 4°C. Centrifugation was carried out at 13,000 rpm for 15 min and the resultant supernatant was collected. Bradford assay was used to quantify the extracted proteins. For western blotting, the membrane was blocked with 2.5% BSA in 1X TBS. Antibodies against GlVps46a (1:8000 dilution) and GlVps46b (1:5000 dilution) was used. As loading control, anti-3-PGK (Invitrogen) was used at 1:8000 dilution. The membranes were incubated with respective antibodies for 2 h. Membranes were washed thrice with 1X TBST and twice with 1X TBS. Anti-mouse or anti-rabbit AP-conjugated secondary antibody (Abcam) was used at 1:8000 dilution in 1X TBS for 1 h. Membranes were washed as previously described and developed using NBT/BCIP.

## Immunolocalization

*Giardia* cells were harvested by chilling the tubes on ice for 5 min, followed by centrifugation at 1000g for 10 min. The cells were washed twice with 1X PBS and fixed with 4% formaldehyde for 15 min. Next, the cells were collected by centrifugation and treated with 0.8 M glycine in 1X PBS for 5 min at room temperature. Cells were harvested by centrifugation and permeabilized by 0.2% Triton X-100 solution, for 8 min. Next, the cells were blocked with 2% BSA for 1.5 h, harvested by centrifugation, and then incubated at 4°C overnight with primary antibody solution for GlVps4a (1:5000), GlVps4b (1:5000), GlVps4c (1:5000), GlVps46a (1:8000) or GlVps46b (1:5000) in 1X PBS. Cells were collected by centrifugation and washed thrice with 1X PBS, followed by incubation with a secondary antibody (anti-rabbit Alexa-488, 1:8000 dilution in 1X PBS) for 2 h. Then the cells were washed thrice with 1X PBS and resuspended in mounting medium containing p-phenylenediamine as a quencher. The sample was imaged with the 63X oil immersion objective of Leica confocal microscope. All incubation was carried out at room temperature unless mentioned otherwise. For colocalization studies, the cells were incubated with primary antibodies against GlVps4a, GlVps46a, or GlVps4c in combination with anti-Glα-SNAP$_{10856}$ (1:5000). Alexa Fluor 488-conjugated anti-rabbit and Alexa Fluor 594-conjugated anti-mouse secondary antibodies were used for detection. Colocalization of GlVps4a, GlVps4b, GlVps4c, and GlVps46b with α-tubulin was performed using anti-α-tubulin (1:5000) (α-tubulin antibody (DM1A), Thermo Scientific) along with the respective rabbit primary antibodies against GlVps proteins. Alexa 594-conjugated anti-mouse secondary antibody was used for α-tubulin detection, while Alexa 488-conjugated anti-rabbit was used for GlVps proteins where applicable.

## Yeast two-hybrid analysis

Constructs with genes cloned in pGBT9 (*TRP1*) or pGAD424 (*LEU2* selection marker), as described above, were transformed into PJ69-4A yeast strain (Takara). To monitor growth on selective dropout plates, equal numbers of cells were spotted, and incubated for 3 days at 28°C. Quantitative estimations of various binary interactions were evaluated through analyses of β-galactosidase activity [34]. The results presented here are the average of three independent experiments, with each experiment comprising three technical replicates per sample. A two-tailed, unpaired t-test was conducted using GraphPad Prism 5 to determine statistical significance.

## Protein expression and co-purification

*E. coli* BL21(DE3) cells were co-transformed with His tagged GlIst1 in pET-24dHisTEV and pETDuet-1 containing either GlVps46b or GlVps4b$_{1-191}$. Protein expression was induced with 0.5 mM IPTG for 4 h at 37°C. Cells were harvested by centrifugation at 8,000 rpm for 5 min and lysed via sonication. Following sonication, cell debris were removed by centrifugation at 10,000 rpm for 30 min at 4°C. The resulting supernatant was loaded on Ni$^{2+}$-NTA affinity column. The column was washed with lysis buffer containing 10 mM and 20 mM imidazole. Bound proteins were eluted with lysis buffer containing 200 mM imidazole. The eluted fraction was analyzed by SDS-PAGE and western blot was performed using anti-His and anti-GlVps46b/GlVps4b antibodies.

## Reverse Transcriptase - PCR

Total RNA was extracted from trophozoite and cysts of Assemblage A isolate WB using TRIZOL (Invitrogen) according to the manufacturer's protocol [31]. cDNA was prepared from total RNA using Reverse Transcriptase (Invitrogen). The PCR reaction was performed with primers (S1 Table) corresponding to the internal sequence of the *glist1* using the following conditions: initial denaturation at 95°C for 5 min, 30 cycles of amplification (95°C for 1 min, 55°C for 1 min, 72°C for 1 min), followed by post extension at 72°C for 10 min.

## Bioinformatic analysis

Sequences of GlIst1, ScIst1, ScVps4, GlVps4b and HsVps4B were obtained from UniProt. The predicted tertiary structures were obtained from AlphaFold, InterPro and Pfam were used for the domain analysis [35–37]. Pairwise sequence alignment was performed using EMBOSS Needle, to calculate the identity and similarity between different ESCRT proteins. S3 Table contains the list of the Gene ID and UniProt ID of the proteins utilized in domain analysis and structure prediction.

## Live cell imaging

BY4742 cells co-expressing either GFP-GlIst1 and ScIst1-RFP, or GFP-GlIst1 and ScVps27-RFP were grown in a synthetic dropout medium until the O.D$_{600}$ reached 0.5-0.6. Cells were harvested by centrifugation at 5000 rpm for 3 min. The supernatant was discarded, and the pellet was re-suspended in YCM. 2 μl of the cells were mounted on a slide, and the sample was imaged with the 63X oil immersion objective of Leica confocal microscope.

## LC-ESI-MS/MS

250 ml of confluent trophozoite culture was incubated on ice for 30 min prior to collection via centrifugation at 1000g for 10 min. The harvested cells were then washed with 1X PBS at 1000g for 10 min and were subsequently resuspended in 200 μl of lysis buffer comprising 50 mM Tris pH 8.0, 120 mM NaCl, 5 mM EDTA, 1% Triton X-100, and a protease cocktail inhibitor (Sigma P8215). Cells were disrupted via sonication and the lysate was obtained by centrifugation at 13000 rpm for 10 min. The lysate was subsequently lyophilized. The lyophilized protein was then resuspended in 20 μl of 100 mM NH$_4$HCO$_3$ (Hi-Media). The protein concentration of the lysate was determined using Bradford assay, and the sample for mass-spectrometry was prepared using 2.5 μg/μl protein. Each sample was combined with 25 μl trifluoroethanol (SRL) and 1.25 μl 200 mM DTT (Roche) and incubated at 60°C for 1h. Subsequently, 5 μl of 200 mM iodoacetamide (Sigma) was added, followed by incubation in the dark for 90 min. Then 1.25 μl of 200 mM DTT was added to the sample and incubated for 1 h. The final step involved the addition of 219 μl water and 100 mM NH$_4$HCO$_3$ to the sample. Overnight digestion of the sample was performed using 2.5 μl of 1 μg/μl trypsin (Promega V528A) at 37°C. The next day, the samples were centrifuged at 13000 rpm for 10 min, and the clear supernatant was collected. Lastly, 1 μl of 0.1% formic acid was added to the supernatant prior to loading the sample for LC-ESI-MS/MS (Waters Corporation). The data analysis was done by using three biological replicates. Progensis QI software was utilized for data acquisition and processing during

LC-MS/MS analysis. The mass spectra of eluted peptides were recorded by the software. To identify peptides, database searches were conducted against known protein sequence information from the UniProt database. In the mapping process, only peptide fragments occurring at least three times were considered. The protein sequence coverage for GlIst1 was 55.8%.

## Results

### Subcellular distribution of the GlVps4 paralogs in trophozoites and during encysting

In previous reports, we have documented the distribution of GlVps4a and GlVps46a using polyclonal antibodies raised against these proteins [25,31]. In trophozoites, GlVps4a predominantly localizes to the PVs located both along the cell periphery and those also around the bare zone surrounding the basal bodies [25]. The distribution of GlVps4a to PVs is supported by the substantial colocalization of this protein with α-SNAP$_{10856}$, a protein known to be present at the PVs (Fig 1A) (S1 Fig) [38]. To further investigate its cellular localization under encysting conditions, we examined encysting trophozoites, both at 16 h and 48 h post-induction (Fig 1A). At 16 h post-induction, cells are expected to have crossed the encystation commitment threshold, thereby activating the molecular machinery that drives morphological changes [39]. At this timepoint, GlVps4a was distributed along the cell periphery (Fig 1A). In addition, a minor pool of the protein was also detected at the cytoplasmic axonemes of the anterior flagella (AF) (Fig 1A). At 48 h post-induction, where further morphological changes occur, there was increased signal intensity at the flagella axonemes (Fig 1A), which were also stained using anti-tubulin antibody (S1 Fig). We also detected the presence of minor pools of GlVps4a at the axonemes of the caudal and posterolateral flagella (CF and PF) pairs (Fig 1A). The protein was also present at the marginal plate, a unique structure just above the AF (Fig 1A). It was also present along the edge of the flanges, extending from both sides of the cell. The extension of the flange membranes is one of the key features of encysting trophozoites, which contributes to changes in cell morphology from the pear-shaped trophozoite to the ovoid cyst [39]. Since the ESCRT machinery is known to induce membrane curvature [40], the localization of GlVps4a along the deforming membrane during encystations may contribute to the membrane sculpting needed for the change in cellular morphology.

Next, we examined the cellular distribution of GlVps4b (Fig 1B) and GlVps4c (Fig 1C). In trophozoites, besides a cytoplasmic pool, GlVps4b was also present at the periphery of the VD and in the membrane-enclosed axonemes of all four types of flagella, structures that were counterstained with anti-tubulin antibody (S1 Fig). It may be noted that, unlike GlVsp4a, GlVps4b is absent at the PVs, while its localization to the cytoplasm is consistent with the distribution of Vps4 orthologue in humans [41]. Upon induction of encystation, while the cytoplasmic signal for GlVps4b persisted, it was absent from the VD periphery or the membrane-encased flagella axonemes. Instead, it was observed at the cytoplasmic axonemes of the PF, where the signal intensified with the progress of encystation (Fig 1B, compare 16 h and 48 h).

Curiously, the distribution of GlVps4c was the opposite of GlVps4b in association with the flagella axonemes. In trophozoites, GlVps4c was present at the cytoplasmic axonemes of the AF, PF, and CF (Fig 1C). Its signal was also detected at the membrane-bound axonemes of the ventral flagella (VF), PF and CF. The signals at these locations colocalized with the signal for tubulin (S1 Fig). Notably, signal enrichment was observed at the tips of all four types of flagella. The protein was also detected at the marginal plate and the perinuclear region. 16 h after induction of encystation, the signal at the cytoplasmic axonemes was lost, except for the AF, where the signal intensity was decreased compared to trophozoites (Fig 1C). The major pool of the protein was at the median body, which is another microtubule-based structure in *Giardia* and is considered to be a reservoir of this cytoskeletal element (Fig 1C) [42]. The signal at the perinuclear region persisted at 16 h but was significantly diminished 48 h after encystation induction (Fig 1C). At this later time, while the signal at the median body persisted, the protein was present in punctate structures at the marginal plate region, and there was also a minor pool at the PVs, a distribution that was confirmed by colocalization with α-SNAP$_{10856}$ (S1 Fig). While the sub-cellular distribution of the three paralogues of GlVps4 are diverse and distinct in trophozoites and encysting cells, they mostly localize to either microtubule-rich structures or at the sites of membrane deformation.

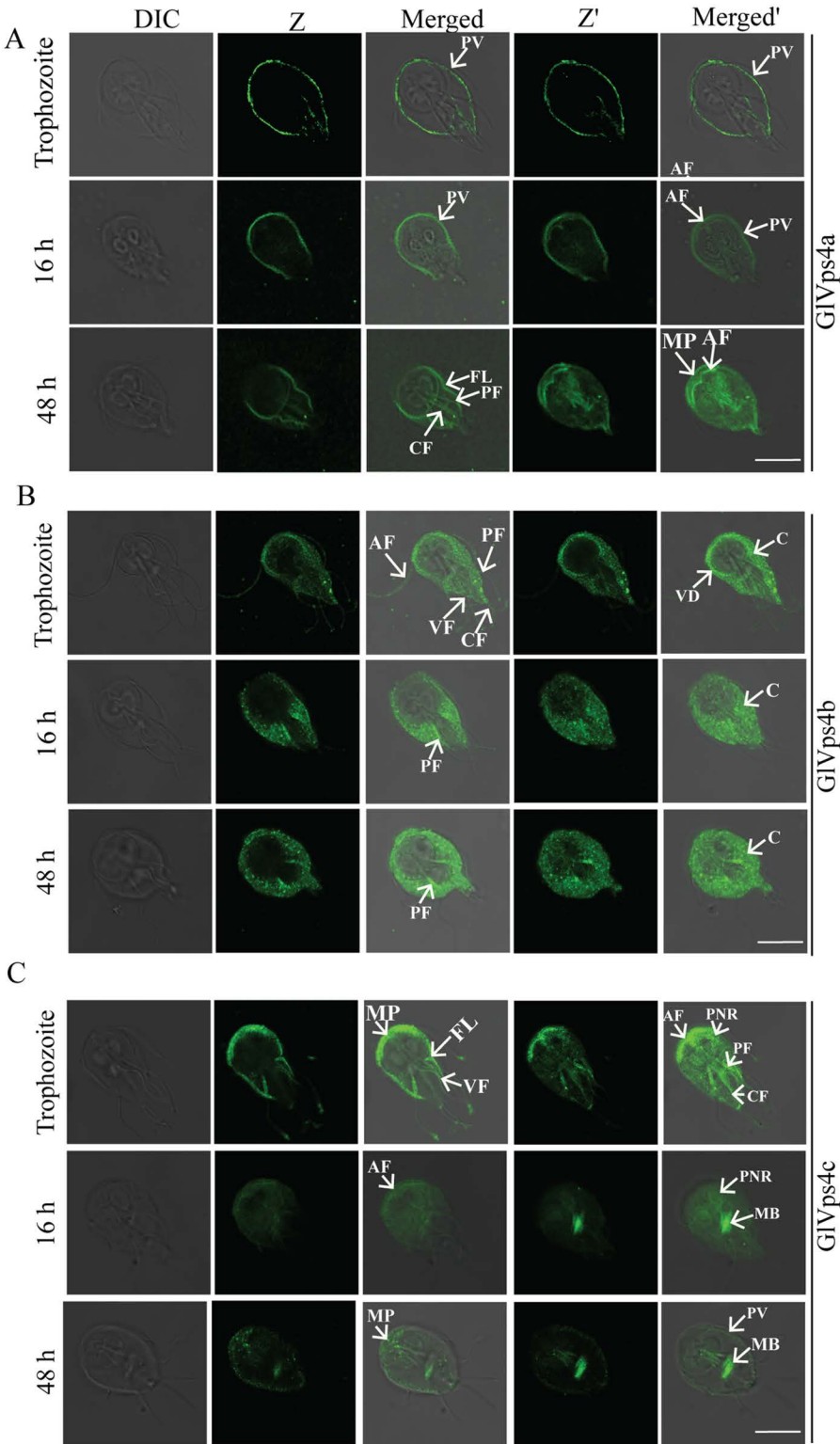

**Fig 1. Sub-cellular distribution of the paralogs of GlVps4 in trophozoites and after induction of encystation.** (A) Immunolocalization of GlVps4a in trophozoites (upper panel) and 16 h (middle panel) and 48 h (lower panel) post-induction of encystation. (B) Immunolocalization of GlVps4b in trophozoites (upper panel), 16 h (middle panel), and 48 h (lower panel) post-induction of encystations. (C) Immunolocalization of GlVps4c in trophozoites

(upper panel), 16 h (middle panel), and 48 h (lower panel) post-induction of encystations. Arrows indicate the following cellular structures: PV: Peripheral Vesicles; AF: Anterior Flagella; PF: Posterior Flagella; VF: Ventral Flagella; CF: Caudal Flagella; VD: Ventral Disc; C: Cytosol; MB: Median Body; MP: Marginal Plate; FL: Flange; PNR: Perinuclear Region. Z and Z' are images of the same frame captured from different focal planes. Scale bar: 8 µm.

## Sub-cellular distribution of the GlVps46 paralogs in trophozoites and during encystation

Next, we investigated the distribution of the two Vps46 paralogs, GlVps46a and GlVps46b. The subcellular distribution of these paralogues, which share 75.8% identity and 85.8% similarity (see Materials and Methods), indicated that while there was some overlap in their distribution, each was also localized to unique regions (Fig 2A and 2B). Both paralogues were detected at the PVs, the cytoplasm, VD periphery and the flagella axonemes (Fig 2A and 2B) (S1 Fig). For GlVps46a, the diffused cytoplasmic signal diminished considerably after 16 h of encystation. Instead, the protein was observed to have a punctate distribution (Fig 2A). GlVps46a was also enriched at the ventrolateral flanges in trophozoites; the signal was also detected at this location at 16 h and 48 h post-induction of encystation. During encystation, the protein was also present as small puncta along the membrane-bound axonemes of the PF and CF. The signal at the VD periphery became more intense in 48 h encysting trophozoites and along the flanges (Fig 2A). For GlVps46b, in addition to the distribution mentioned above, it was enriched in the flagellar pores of the AF and PF and similar to GlVps4c, it was also observed at the flagellar tips (Fig 2B). Incidentally, none of the other ESCRT components we have studied displayed such distribution at the flagella pores. During encystation, while the axoneme signal persisted in 16 h encysting cells, the pool of GlVps46b at the flagellar tip and the flagellar pores was lost (Fig 2B). At 48 h, the axoneme signal too was lost.

Both GlVps46a and GlVps46b were present along the edges of the VD, but their distribution pattern during encystation was not identical; while the signal for GlVps46a increased with the progress of encystation (Fig 2A), that of GlVps46b was detected at the VD periphery of trophozoites and 16 h trophozoites but was absent in 48 h encysting trophozoites (Fig 2B) (S1 Fig). The VD is very important for the survival of this parasite within the host gut [43]. The structure of this appendage involves a tight wrapping of the plasma membrane along its margins. Such sharp membrane bending involves the induction of negative membrane curvature (bending of the membrane away from cytoplasm), and presently, only the ESCRT machinery is known to be capable of such membrane bending. GlVps46b was also detected at the overlap zone of the VD in trophozoites and the signal at this location was more pronounced in 16 h encysting cells (Fig 2B). The signal intensity diminished considerably, but it was still evident in 48 h encysting cells. Curiously, a substantial pool of GlVps46b was also present in the basal body at 16 h encysting trophozoites. Based on the observations described above, we conclude that both the paralogues of GlVps46 are associated with the unique cellular appendages that define this organism. In particular, they localize to sites where membranes bend sharply, such as the tips of flagella, flanges and VD periphery. Their re-localization during encystation also underscores their importance in bringing about morphological transition during encystation.

The distribution of the GlVps4 and GlVps46 paralogues in trophozoites and during encystation are summarized in S4 Table. The results of the localization study showed that on various occasions, the paralogous proteins Vps4 and Vps46 of *G. lamblia* were detected in the same subcellular location. However, there are also locations where these two sets of paralogues are present alone.

## Presence of an Ist1 orthologue in *Giardia lamblia*

Studies in yeast show that ScVps46 regulates the recruitment of ScVps4 to the endosomal surface [44]. We wanted to determine if there is a selective interaction between the paralogues of GlVps4 and GlVps46. Direct contact between ScVps46 and ScVps4 is well documented in yeast, and this interaction can be detected with the yeast two-hybrid (Y2H) assay [45]. Using the same assay, we failed to detect any interaction between the two paralogues of GlVps46 and the three paralogues of GlVps4, even though the yeast proteins showed high-affinity interaction (S4 Fig). Since both Vps46

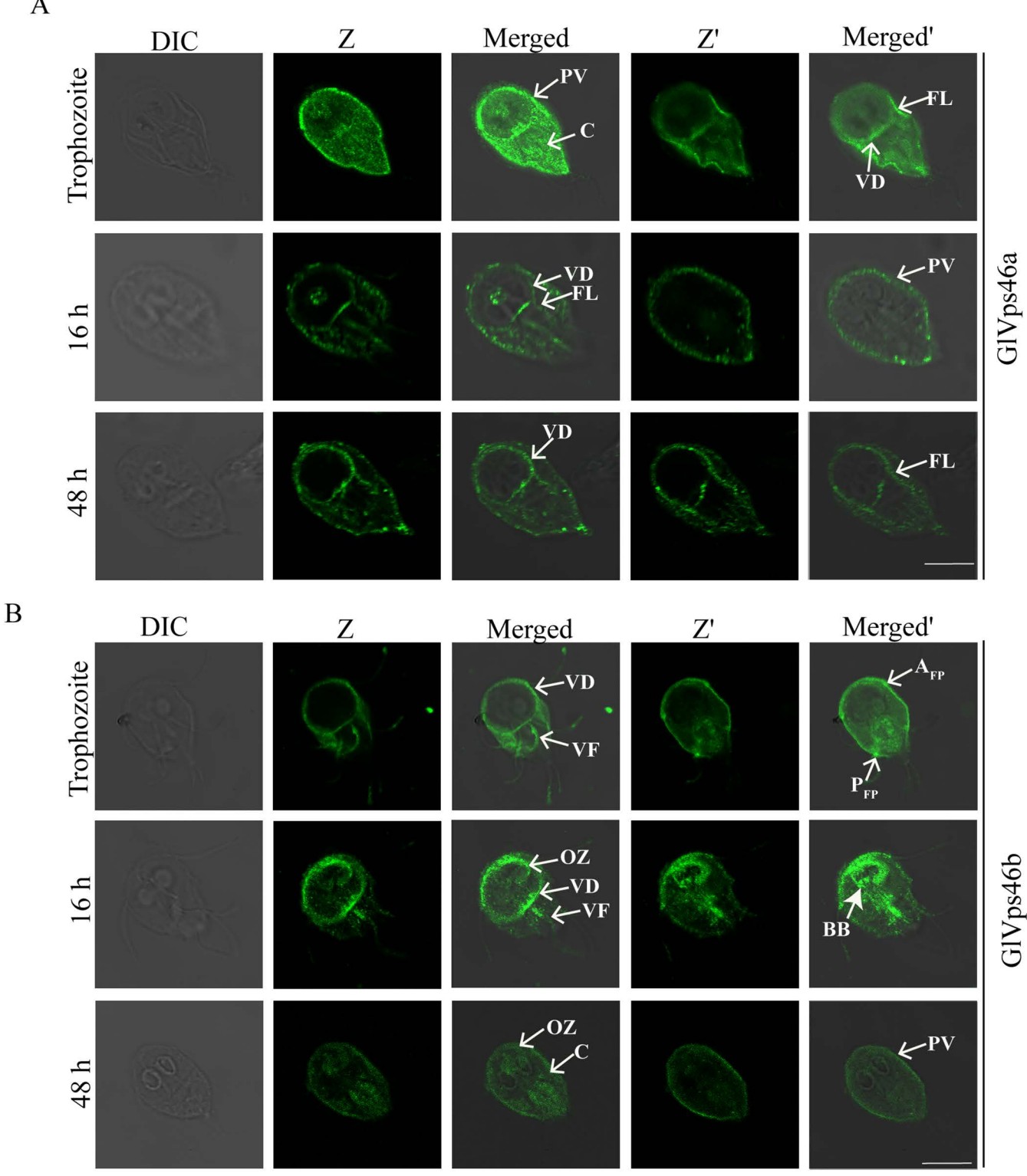

**Fig 2. Subcellular distribution of the paralogs of GlVps46 in trophozoites and after induction of encystation.** (A) Immunolocalization of GlVps46a in trophozoites (upper panel), 16 h (middle panel) and 48 h (lower panel) post-induction of encystations. (B) Immunolocalization of GlVps46b in trophozoites (upper panel), 16 h (middle panel) and 48 h post-induction of encystation (lower panel). Arrows indicate the following cellular structures: PV:

Peripheral vesicles; C: Cytosol; A$_{FP}$: Anterior Flagellar Pore; P$_{FP}$: Posterior Flagellar Pore; FL: Flange; VD: Ventral Disc; VF: Ventral Flagella; OZ: Overlap Zone; BB: Basal Body. Z and Z' are images of the same frame captured from different focal planes. Scale bar: 8 µm.

and Vps4 play important roles in ESCRT-mediated membrane deformation, we hypothesized that although the *Giardia* orthologues do not interact directly, they may be brought into close apposition onto the membrane surface via their interactions with other ESCRT components. Studies in yeast and metazoans show that Ist1, a key ESCRT-III auxiliary component, interacts with both Vps4 and Vps46 [19,46,47]. A search of the *Giardia* reference genome (Assemblage A isolate WB) indicated the presence of a putative GlIst1 (Gl50803_11129). Such putative GlIst1 orthologues were also encoded in the genomes of Assemblage B isolate GS (GL50581_3698) and Assemblage E isolate P15 (GLP15_1481) (Fig 3A). The ORF is also present in the recently sequenced genome of strain BAH15c1of Assemblage B (QR46_3855). However, it was not detected in the other sequenced *Giardia* genomes. Using reverse transcription PCR, we confirmed the expression of the *glist1* in both the trophozoite and cyst stages of Assemblage A isolate WB, indicating that a GlIst1 is produced in *Giardia* (Fig 3B)

### GlIst1-GlVps4b interaction is mediated by the MIM and MIT domain

If GlIst1 is functionally analogous to the yeast orthologue, it is likely to interact with some or all of the paralogues of GlVps4 and/or GlVps46. Using the Y2H assay, we first determined if GlIst1 could physically interact with the GlVps4 paralogues. The known interaction between the yeast orthologues, ScIst1 and ScVps4, served as the positive control. Based on the quantitative estimation of the β-galactosidase activity, we also observed that the expression of the LacZ reporter was turned on when cells co-expressed the Gal4 DNA binding domain fused to ScIst1 (BD-ScIst1) and the Gal4 activation domain fused with ScVps4 (AD-ScVps4) (Fig 4A). We detected β-galactosidase activity when BD-GlIst1 was co-expressed with the AD-GlVps4b, but not when the former was co-expressed with either AD-GlVps4a or AD-GlVps4c, indicating selective binary interaction between GlIst1 and GlVps4b. A comparison of β-galactosidase activity indicates that this interaction within this *Giardia* protein pair was stronger than between the yeast orthologues BD-ScIst1 and AD-ScVps4 (Fig 4A). Additionally, there was no interaction between BD-ScIst1 and AD-GlVps4b or between BD-GlIst1 and AD-ScVps4, indicating that Ist1 and Vps4 orthologues only interact with proteins from the same organism. Since the *Giardia* orthologues of both these proteins significantly diverge from their yeast counterparts, it may be possible that the nature of the interaction between GlIst1 and GlVps4b may be different from that in yeast.

Analyses of the interaction between Vps4 and Ist1 orthologues from multiple organisms show that it is mediated by the N-terminally-located MIT domain of Vps4 and the C-terminal MIT Interacting Motif (MIM) of Ist1 [19,48]. This interaction occurs via MIM1, wherein the MIM alpha helix binds to the groove formed by the second and third alpha helices of the MIT domain [49]. We analyzed the sequence of GlVps4b and GlIst1 to determine if they have MIT and MIM, respectively. Interestingly, while the sequence of GlVps4b is 50.2% similar to ScVps4, none of the domain prediction databases hosted by InterPro recognized an MIT domain towards the N-terminus of the *Giardia* protein, even though the presence of the AAA ATPase domain and the Vps4_C domain was recognized. However, an MIT-like domain must be present in GlVps4b as it can functionally substitute for ScVps4 [25]. Consistently, AlphaFold indicates the presence of a three-helix structure at its N-terminus that is structurally similar to the MIT domains of ScVps4 and VPS4B in humans (Fig 4B). These three helices span residues 30–109 of the protein (Fig 4C). Based on this, we hypothesized that a non-canonical MIT domain is present at the N-terminus of GlVps4b.

To detect if such a non-canonical MIT domain is present in GlVps4b, we carried out co-purification assay. *E. coli* cells were co-transformed with constructs expressing GlIst1 with a 6x His tag and the N-terminal fragment of GlVps4b that includes all residues prior to the AAA ATPase domain, GlVps4b$_{1-191}$. Cells expressing only GlVps4b$_{1-191}$ served as control. Purification of GlIst1 via immobilized metal affinity chromatography revealed the enrichment of the GlVps4b fragment in

A

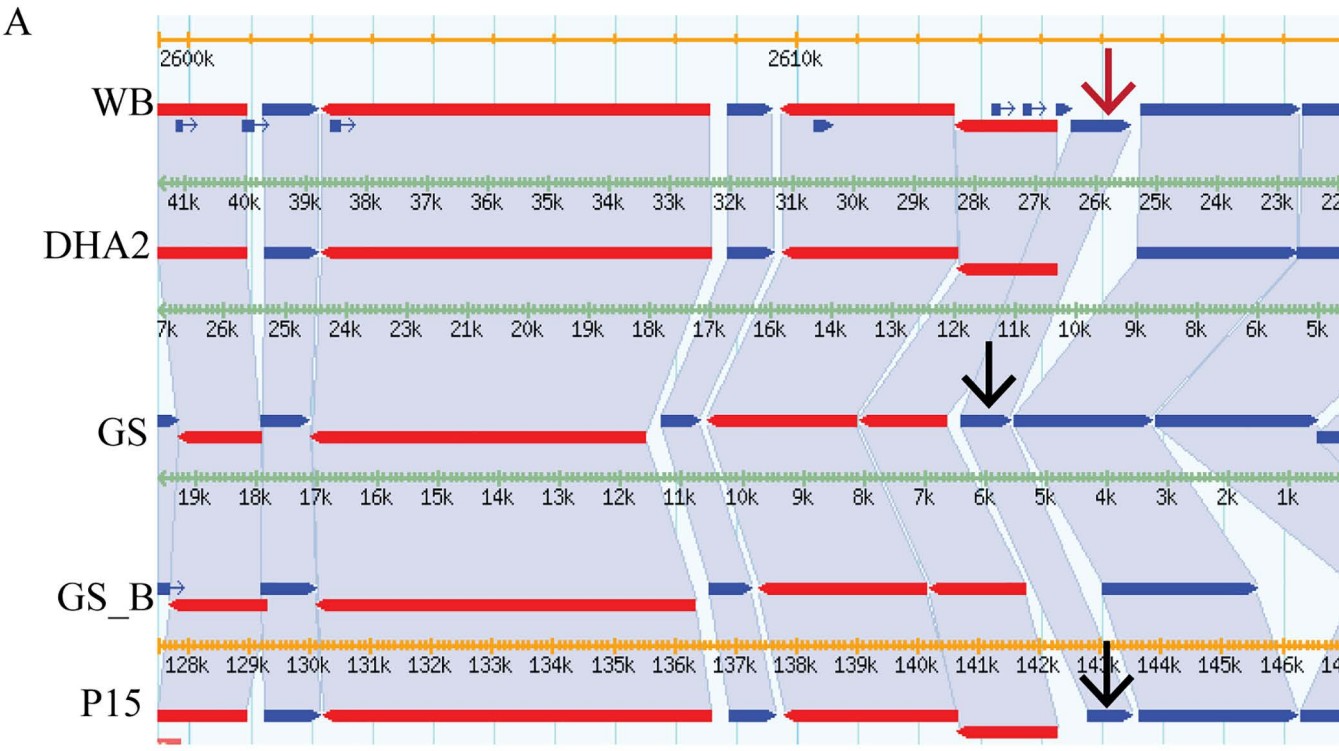

B

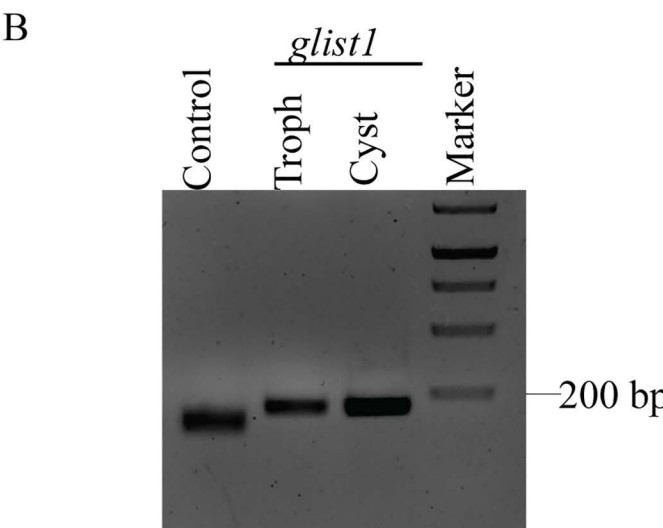

—200 bp

**Fig 3. Synteny and expression of glist1.** (A) Synteny of *glist1* in different isolates of *Giardia*. The red arrow indicates the location of *glist1* in the WB isolate, while the black arrows mark its position in other *Giardia* isolates. (B) Reverse transcription PCR to detect the expression of *glist1* in trophozoites and cysts. Expression of *glvps46b* serves as a positive control.

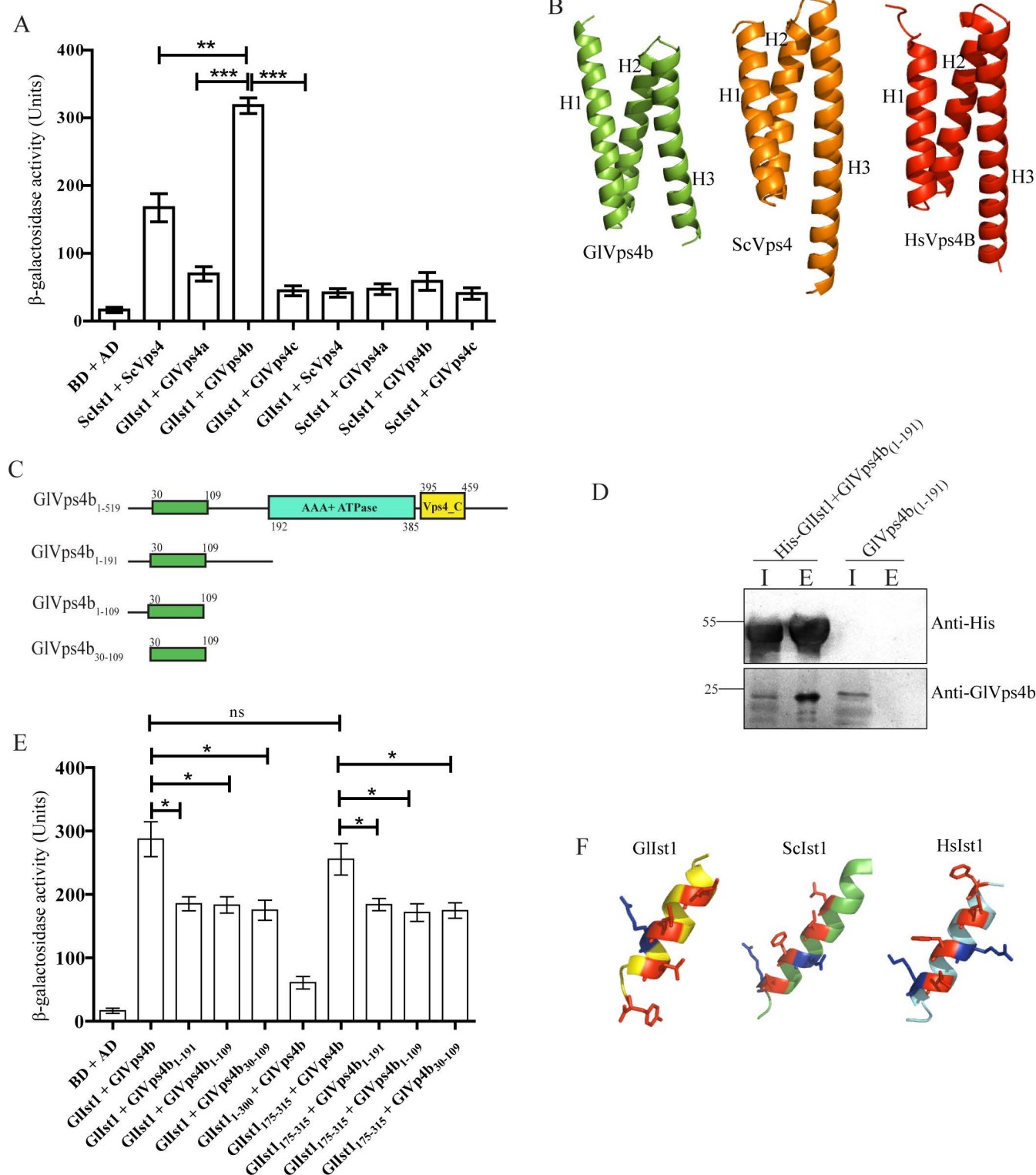

**Fig 4. MIT domain of GlVps4b interacts with GlIst1.** (A) β-galactosidase activity of PJ69-4A cells transformed with different constructs expressing fusion proteins featuring either the Gal4 DNA binding domain (BD) or its activation domain (AD). The expression of BD and AD alone served as negative control. (B) AlphaFold predicted secondary structure of GlVps4b, ScVps4 and HsVps4B. H1: Helix 1, H2: Helix 2 and H3: Helix 3 of the predicted

three-helix bundle comprising the MIT domain. (C) Domain architecture of full-length GlVps4b or its different fragments used in the assay. (D) Pull-down assay showing copurification of GlVps4b$_{1-191}$ with 6x His-tagged GlIst1. SDS-PAGE followed immunoblotting with anti-GlVps4b or anti-His antisera indicated that the presence of GlVps4b$_{1-191}$ in the elution fraction (E) was dependent on the co-expression of the 6x His-tagged GlIst1 in *E. coli*. Sample loading in input (I) lanes is 3X of that loaded for E. (E) β-galactosidase activity of PJ69-4A cells transformed with different constructs expressing full-length or truncated versions of GlVps4b and GlIst1. Statistical significance: ns- not significant (p > 0.05), *-significant (p ≤ 0.05), **-very significant (p ≤ 0.01), and ***-extremely significant (p ≤ 0.001). (F) AlphaFold predicted secondary structure of the putative MIM of GlIst1 and the canonical MIM structural elements of ScIst1 and HsIst1.

the elution fraction when compared to the total supernatant (Fig 4D) (S5 Fig). In the negative control, the GlVps4b$_{1-191}$ fragment was not detected in the elution fraction, which indicates that its presence in the eluant is dependent on its copurification with GlIst1.

Next, we used two-hybrid analysis to compare if there is any difference in the interactions of GlIst1 with either the full-length GlVps4b or GlVps4b$_{1-191}$. We used β-galactosidase activity to quantitatively compare these two interactions, and the results indicate that while there was some reduction in the affinity between the GlVps4b$_{1-191}$ fragment and GlIst1, the decrease was not substantial as the p-value was < 0.05 (Fig 4E). This indicates that interaction between GlVps4 and GlIst1 is mediated by the N-terminal segment of the former.

Secondary structure prediction of GlVps4b indicates that a short alpha helix precedes the three-helix bundle, followed by a largely unstructured region and finally another short alpha helix (Fig 4C). To determine if the three-helix bundle constitutes a non-canonical MIT, we carried out further truncations from the C-terminal end so that GlVps4b$_{1-109}$ retains only up to the three-helix bundle. The β-galactosidase activity was nearly identical when either GlVps4b$_{1-191}$ or GlVps4b$_{1-109}$ interacted with GlIst1 (Fig 4E). Further truncation from the N-terminus generated the fragment GlVps4b$_{30-109}$ that retained only the triple helix bundle. Even this fragment exhibited an almost similar affinity for GlIst1 as indicated by β-galactosidase activity which was similar to those of the previous two fragments (Fig 4E). These results indicate that residues spanning 30–109 of GlVps4b are sufficient to mediate the interaction between this GlVps4 paralogue and GlIst1. Also, as these residues are predicted to form a three-helix bundle, GlVps4b likely contains a non-canonical MIT domain that cannot be recognized by any of the commonly used domain prediction databases.

Since GlVps4b's interaction with GlIst1 is mediated via an MIT domain, this domain likely interacts with a MIM in GlIst1. AlphaFold predicts the presence of a putative MIM-like amphipathic helix from residues 302–315 (Fig 4F). Similar to the canonical MIMs in the yeast and human orthologues, this helix is located near the C-terminal end of GlIst1, and it contains both hydrophobic (I305, L309, L312, and Y315) and positively charged (R310) residues (Fig 4F). Although Y315 is located in an unstructured region that is very close to the putative alpha helix, it may be induced into adopting an alpha-helical conformation upon binding to GlVps4b. To assess whether this region mediates interaction with GlVps4b, we generated a truncation mutant, GlIst1$_{1-300}$, fused with the BD. Unlike the full-length GlIst1, this mutant did not interact with the AD-GlVps4b as evidenced by the β-galactosidase activity (Fig 4E). Notably, the interaction was restored when the C-terminal portion of GlIst1, GlIst1$_{175-315}$, was expressed as indicated by β-galactosidase levels that were almost equal to those cells expressing the full-length GlIst1 and AD-GlVsp4b (Fig 4E). The interactions of GlIst1$_{175-315}$ with those fragments of GlVps4b mentioned above were almost similar to that of the full-length GlIst1. Taken together, it may be inferred that despite the absence of the canonical MIT and MIM in GlVps4b and GlIst1, respectively, the MIT-MIM interaction appears to have been retained between GlIst1 and GlVps4b in this parasite.

## GlIst1 interacts only with GlVps46b

Next, we probed if GlIst1 can interact with the GlVps46 paralogues using yeast two-hybrid analysis. β-galactosidase activity measurements indicated that there is a strong interaction between BD-ScIst1 and AD-ScVps46, which is consistent with previous reports (Fig 5A) [50]. Of the two Vps46 paralogues of *Giardia*, only GlVps46b could interact with GlIst1, and

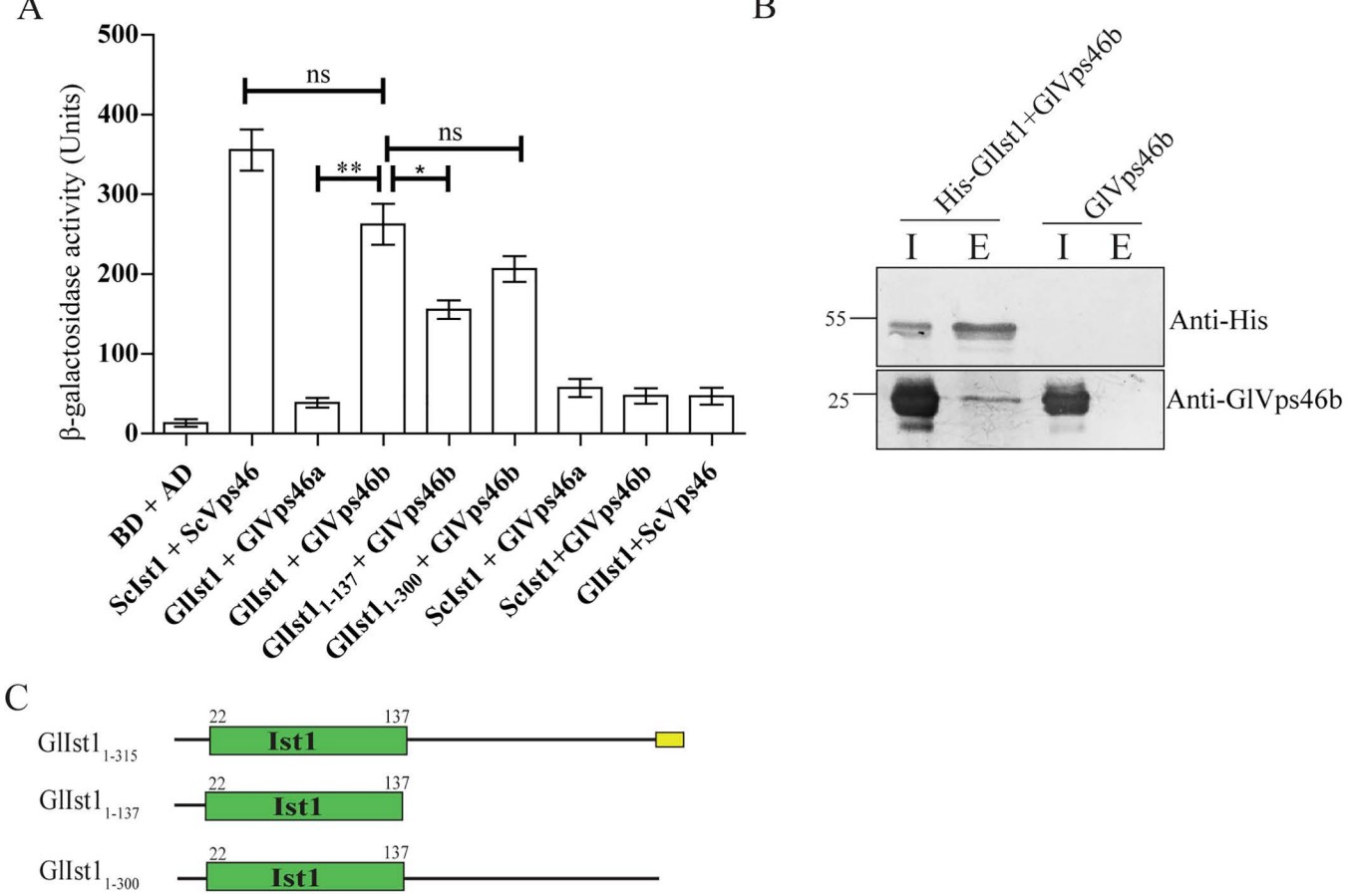

**Fig 5. Ist1 domain of GlIst1 interacts with GlVps46b.** (A) β-galactosidase activity of PJ69-4A cells transformed with different constructs expressing full-length or truncated versions of GlVps46b and GlIst1 fusion proteins featuring either the Gal4 DNA binding domain (BD) or its activation domain (AD). The expression of BD and AD alone served as negative control. (B) Pull-down assay showing copurification of GlVps46b with 6x His-tagged GlIst1. SDS-PAGE followed immunoblotting with anti-GlVps46b or anti-His antisera indicated that the presence of GlVps46b in the elution fraction (E) was dependent on the co-expression of the 6x His-tagged GlIst1 in *E. coli*. Sample loading in input (I) lanes is 3X of that loaded for E. (C) Domain architecture of full-length GlIst1 or its different fragments used in the assay. The Ist1 domain is depicted in green, and the MIM, spanning residues 302–315, is shown in yellow.

the quantification of the β-galactosidase activity indicates that the affinity between this *Giardia* protein pair was comparable to that observed for the orthologous protein pair from yeast. Similar to the species specificity observed in the case of the Ist1-Vps4 interaction described previously, here, too, we observed that there was no interaction between ScIst1 and GlVps46b or GlIst1 and ScVps46 (Fig 5A). This again underscores the divergence of the late-ESCRT proteins of *Giardia* from those of yeast.

The interaction of GlIst1 with GlVps46b was validated using copurification. Expression of 6x His-tagged GlIst1 and GlVps46b, followed by capture on Ni²⁺ NTA beads shows the presence of GlVps46b in the elution fraction (Fig 5B) (S5 Fig). Just like GlVps4b, the band of GlVps46b was absent when GlIst1 was not co-expressed. This observed copurification indicates that GlIst1 and GlVps46b interact with each other.

The interaction between Vps46 and Ist1 in higher organisms, including yeast and humans, is mediated by the N-terminal region of Ist1, which includes the Ist1 domain, and the MIM located at the C-terminus of Vps46 [47]. Sequence

analysis revealed that GlIst1 shares a low level of similarity with its yeast and human orthologues, at 19.2% and 27.7%, respectively. Despite these low values, domain analyses revealed the presence of an Ist1 domain, spanning residues 22–137, in GlIst1 (Fig 5C). At 116 residues, this predicted Ist1 domain is considerably smaller than the corresponding domain in the yeast and human orthologs, which are 164 and 171 residues long, respectively. Pairwise alignment also reveals that the identified Ist1 domain of GlIst1 shares only 11.6% identity and 22.2% similarity with the domain in the yeast ortholog and 15.0% identity and 25.2% similarity with the domain in the human orthologue. Besides these low identity values, the reduced size of the Ist1 domain in GlIst1, further indicate its non-canonical nature. Hence, we were curious to determine if this smaller non-canonical Ist1 domain can mediate the interaction with GlVps46b. While the β-galactosidase activity of transformants co-expressing BD-GlIst1$_{1-137}$ and AD-GlVps46b was significantly higher than the corresponding negative control, it was lower than that observed with the full-length GlIst1 (Fig 5A). This suggests that additional segments of the GlIst1 protein are likely to contribute to the interaction with GlVps46b. Consistent with this hypothesis, a previous report suggested, that the α-5 of ScIst1 provides a binding surface for the MIM of ScVps46 [47]. AlphaFold indicates the presence of such a helix outside of the predicted Ist1 domain in GlIst1, spanning residues 172–187. To confirm that sequences beyond the predicted Ist1 domain are required for GlVps46b binding, we created a truncated version of this protein that is missing only the putative MIM helix, GlIst1$_{1-300}$ (Fig 5C). Consistent with our expectation, this larger fragment interacted more efficiently with GlVps46b as the level of β-galactosidase was comparable to that produced with the full-length BD-GlIst1 (Fig 5A). Taken together, the above results indicate that GlIst1 can selectively interact with GlVps46b alone and this interaction is similar to that observed between the corresponding yeast orthologues. However, the interaction surfaces are likely to have undergone evolutionary changes that make each interaction species-specific.

## Difference in cellular distribution of GlIst1 and ScIst1

Beyond the species-specific interactions with Vps4/Vps46, where each Ist1 orthologue binds exclusively to the proteins encoded in its own genome, GlIst1 and ScIst1 may also differ in other functional properties. AlphaFold structures of these two proteins indicate no major deviation in the number of alpha helices, with minor differences in the length of individual helices (S6 Fig). Since an important aspect of Ist1 function in the ESCRT pathway is its recruitment to membranes where the ESCRT pathway operates, we determined if there are any differences in the subcellular distribution between these two orthologous proteins by monitoring the distribution of fluorescently tagged ScIst1 and GlIst1 in yeast. ScIst1-RFP was present in a single punctum within each cell, which is consistent with previous reports of its endosomal localization (Fig 6) [50]. On the other hand, GFP-GlIst1 localization pattern revealed a punctum adjacent to a ring. The punctate signal for GFP-GlIst1 and ScIst1-RFP colocalized indicating that GlIst1 is present in the yeast endosome. This endosomal distribution of GlIst1 was further confirmed by colocalizing it with another well-documented endosomal protein, ScVps27 (ScVps27-RFP) (Fig 6) [51]. The observed ring-like structure, in close apposition to the endosome, is likely to be the vacuolar membrane. This was confirmed by counterstaining with FM4–64, a vital dye that is used to stain the vacuolar membrane, and observing colocalization of the FM4–64 signal with that of GFP-GlIst1 (Fig 6) [52]. Thus, unlike ScIst1, GlIst1 has attributes that enable it to localize to the vacuole membrane, which indicates a functional divergence between the two proteins.

ScIst1 function is known to be regulated by post-translational modifications (PTM) [53]. Hence, we determined the PTM profile of GlIst1 to assess if it undergoes any specific modification, such as myristoylation, palmitoylation or prenylation, that may facilitate interaction with membranes [54]. The proteomic profiling of GlIst1 was carried out using LC-ESI-MS/MS analysis from both *Giardia* trophozoites and 16 h encysting cells. This analysis indicated myristoylation of GlIst1 at K43 position in both samples; no palmitoylation or prenylation was detected (S7 Fig). None of these PTMs have been documented in either ScIst1 or human IST1. Incidentally, similar to documented ubiquitylation of ScIst1 at K135, we detected ubiquitylation of GlIst1 at K33 indicating possible conservation of certain regulation of these orthologues [55]. This suggests that while certain regulatory mechanisms, such as ubiquitination, are conserved, other modifications appear unique

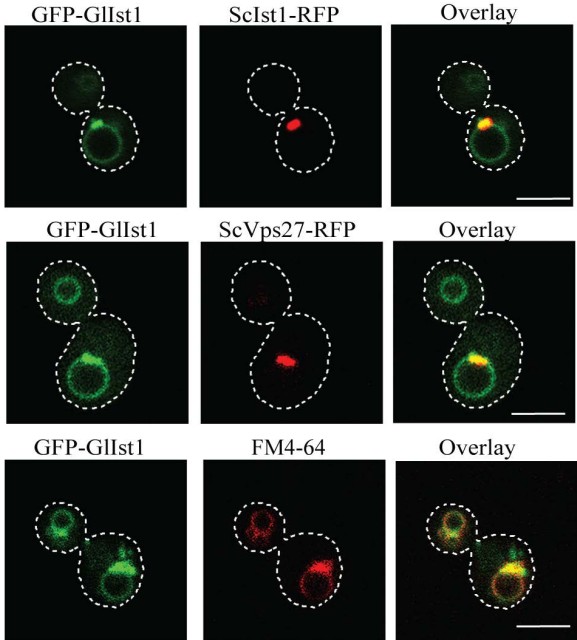

**Fig 6. Difference in subcellular distribution of Ist1 orthologues of yeast and *Giardia*.** Co-localization of GFP-GlIst1 and ScIst1-RFP proteins in yeast (upper panel); co-localization of GFP-GlIst1 with ScVps27-RFP (middle panel) and with FM4–64 (lower panel) Yellow signals in merged images indicate colocalization between the respective fluorescent markers. Scale bar: 8 μm.

to *Giardia*, which may underscore the evolutionary divergence in the functioning of this ESCRT protein between yeast and *Giardia*.

## Discussion

The present study was undertaken to determine if there exists functional divergence(s) between the various Vps4 and Vps46 paralogues in *Giardia*. To answer this, we have elucidated the distribution of all the paralogs of GlVp4 and GlVps46 in both trophozoites and encysting trophozoites as distinct patterns of cellular distribution are indicative of functional divergence. This comprehensive immunolocalization data, obtained without the addition of any external tag that may interfere with incorporation into the ESCRT complex, provides new insight into the potential functions of the ESCRT components within the parasite. Our results show that the late ESCRTs have a diverse cellular distribution and there is a change in the distribution pattern during encystation. This not only indicates functional divergence, but also provides clues regarding the possible cellular roles of these components in *Giardia*.

The ESCRT machinery is known to induce membrane curvature [40]. Studies in model organisms have indicated that ESCRTs are involved in the formation of narrow membranous structures, such as the midbody connecting the two daughter cells just before cytokinesis and the tubular extensions of endosomes [2,14]. In the former case, ESCRT-III components are assembled inside the tubes while in the latter, the ESCRT filaments line their outer surface. Consistently, we have observed that the paralogs of GlVps4 and GlVps46 are primarily present at the sites of acute membrane bending, such as the periphery of VD, the edges of the flanges and the tips of the flagella. All of these structures are important virulence factors as they enable survival of the parasite within the host gut [56,57]. Thus, ESCRT components at membrane protrusions along the edge of the flanges and at sites where there is wrapping of membranes around microtubule structures may be instrumental in bringing about the membrane deformation required at these sites. Presence of the ESCRTs at flagellar tips may also have

other functions. Previous reports indicate that in *Chlamydomonas*, VPS4 is involved in the release of ectosomes from flagella [58]. This study also found that paralogs of GlVps4 and GlVps46 are also observed at different parts of flagella, indicating their possible contribution to flagellar function. Whether *Giardia*, like *Chlamydomonas*, uses extra-cellular vesicles for communication remains an open question. Incidentally, the localization data indicates possible functional segregation in the Vps46 paralogues as GlVps46a is largely confined to sites of acute membrane bending, while GlVps46b also associates with microtubule structures, such as the OZ, the flagella axonemes and the flagella pores (Fig 2A and 2B).

Giardia has a unique cytoskeletal structure, primarily made of microtubules, and interestingly, the ESCRT components investigated in this study are observed in multiple microtubule-rich regions. Besides GlVps46b, the Vps4 paralogues are also associated extensively with microtubule structures. All three of them are present at the cytoplasmic axonemes, even though at some locations, there is no overlap between them (Fig 1). For example, GlVps4b is absent from the cytoplasmic axonemes of the anterior flagella and only GlVps4c is present at the median body. Such association of Vps4 paralogues with microtubule-based structures may be dependent on the MIT domain, which is known to be associated with microtubule-related cellular processes [59].

Besides localizing to sites of acute bending of the plasma membrane and microtubule-based structures, the ESCRT components were also detected at organellar membranes, that of the PVs. These compartments are involved in endocytosis, are crucial for transporting proteins and other macromolecules necessary for the parasite's survival and pathogenicity. Almost all ESCRT components are found in PVs, suggesting an ESCRT-dependent deformation of this compartment's membrane. It has been previously reported functional diversification of PVs based on the shapes of these organelles [60]. Spherical PVs primarily serve as the digestive compartments, whereas tubular or elongated PVs are primarily associated with dynamic exchange processes that facilitate the transport of materials between the PV lumen and the extracellular environment; their elongated shape may enhance their ability to interact with the plasma membrane. Similar endosomal tubule formation at the ER occurs with the help of CHMP1B, along with other ESCRTs and spastin [2]. Our study also showed that GlVps46b is present in PVs, which raises the possibility that it may contribute to the formation of tubular PVs along with *Giardia*'s spastin.

It is intriguing that unlike the multiple paralogues for GlVps4 and GlVps46, GlIst1, another late ESCRT component, appears to be encoded in the genomes of only a subset of *Giardia* species. Studies in yeast indicate that while ScIst1 regulates the recruitment of ScVps4 via ScVps46, *ist1* deletion mutants do not have any observable defects in the ESCRT pathway, indicating that Ist1 plays a redundant role [61]. Such a non-essential role may explain its loss from multiple *Giardia* genomes. Its retention in some genomes may be linked to its ability to serve as a platform for bringing together GlVps4b and GlVps46b on the surfaces of membranes. This also raises the possibility of different modes of membrane recruitment for different Vps4 and Vps46 paralogues within the same *Giardia* species and similar divergences between orthologous components from different species. Along the same lines, there appears to be considerable heterogeneity in the role of PTMs in modulating ESCRT functions. While our results indicate that GlIst1 undergoes myristoylation, such a modification has not been reported for Ist1 of model organisms. Incidentally, in yeast, ScVps20 is recruited to the membrane by myristoylation [62]. Thus, throughout evolution, myristoylation as a means of membrane recruitment of ESCRT-III components may have arisen as an independent event.

Based on the diverse sub-cellular locations to which these proteins are targeted, it is possible that the ESCRT components play roles in multiple cellular processes in *Giardia*. Although some of these functions are similar to those documented in other species, some may have evolved specifically to address the challenges of sustaining the unique biological features of this gut-dwelling organism. Moving forward, our work will focus on functional investigations, including knockdown and other perturbation approaches, to delineate the precise contributions of these late-ESCRT components in morphological stage conversion and lifecycle progression. Also, colocalization of the paralogs will add to our understanding of the cellular roles of these proteins. If these studies indicate extensive roles of the ESCRT pathway in maintenance of parasite morphology and encystation and/or excystation, then the observed species specificity with respect to the interactions between Ist1-Vps4 and Ist1-Vps46 pairs from yeast and *Giardia* may be leveraged for designing new therapeutics.

## Supporting information

**S1 Table. Sequences of primers used in this study.**
(DOC)

**S2 Table. List of constructs used in this study.**
(DOCX)

**S3 Table. List of Gene ID and UniProt ID used in this study.**
(DOCX)

**S4 Table. Summary of the localization of different paralogues of GlVps4 and GlVps46.**
(DOCX)

**S1 Fig. Co-localization of GlVps4/GlVps46 with cellular markers in trophozoites and encysting trophozoites.** (A) GlVps4a (green) with α-SNAP$_{10856}$ (red) in trophozoites. (B) GlVps4a (green) with α-tubulin (red) at 48 h post-encystation. (C) GlVps4b (green) with α-tubulin (red) in trophozoites. (D) GlVps4c (green) and α-tubulin (red) in the trophozoites. (E) GlVps4c (green) with α-SNAP$_{10856}$ (red) at 48 h post-encystation. (F) GlVps46a (green) with α-SNAP$_{10856}$ (red) in trophozoites. (G) GlVps46b (green) with α-tubulin (red) at 16 h post-encystation. Merged images show colocalization in yellow. Scale bar: 8 μm. (TIF)

**S2 Fig. Specificity of antibodies against the paralogs of GlVps4.** (A) Cells treated with pre-immune sera collected from animals prior to immunization with GlVps4b (upper panel) or with GlVps4c (bottom panel). Scale bar 8 μm. (B) Western blotting of the three 6x-His tagged GlVps4 paralogues expressed in *E. coli* and then purified from the bacterial extracts. Blots were incubated with anti-GlVps4a antibody (left), with anti-GlVps4b antibody (middle), and with anti-GlVps4c antibody (right). The detections of bands of sizes ~56kDa, ~60 kDa, and ~30k Da demonstrate the specificity of GlVps4a, GlVps4b, and GlVps4c antibodies, respectively. (C) Western blots were performed using trophozoite extracts. The blot was developed using the anti-GlVps4b antibody (left), while the blot was developed using anti-GlVps4c (right). (TIF)

**S3 Fig. Specificity of antibodies against the paralogs of GlVps46.** (A) Cells treated with pre-immune sera were collected from animals prior to immunization with GlVps46a (upper panel). Cells treated with pre-immune sera were collected from animals prior to immunization with GlVps46b (bottom panel). Scale bar 8 μm. (B) Western blot analysis was performed using PJ69-4A transformants expressing either BD-tagged GlVps46a or BD-tagged GlVps46b. The western blot was performed with anti-GlVps46b antibody (left) and anti-GlVps46a antibody (right). A ~37.4 kDa band (right) and a ~37.8 kDa band (left) were detected. (C) Western blots were performed using trophozoite extracts. The blot was developed using the anti-GlVps46a antibody (left) and anti-GlVps46b antibody (right). All western blots were performed using prestained protein ladder from Thermo Scientific (26616) markers except for GlVps46b (26619). (TIF)

**S4 Fig. Lack of binary interaction between the paralogs of GlVps4 and GlVps46 using yeast two-hybrid assay.** PJ69-4A cells were transformed with various BD and AD fusion combinations, as indicated in the figure. Growth of the PJ694-A transformants were monitored on SD leu⁻trp⁻ (left panel), SD leu⁻trp⁻his⁻ with 2.5 mM 3-AT (middle panel), and leu⁻trp⁻ade⁻ (right panel). The experiment was repeated with constructs expressing proteins with reversal of BD or AD fusion (bottom panels). (TIF)

**S5 Fig. Interaction between GlIst1 and GlVps4b/GlVps46b.** (A) 12% SDS-PAGE showing co-elution of GlIst1 along with GlVps4b$_{1-191}$ in the elution fraction. (B) 12% SDS-PAGE showing co-elution of GlIst1 along with GlVps46b in the

elution fraction, S: total supernatant; FT: flowthrough; E: elution; W: wash. (C) An extended representation of quantification of β-galactosidase activity shown in Fig4A, (D) Fig 4E and (E) Fig 5B.
(TIF)

**S6 Fig. Alpha-Fold predicted structures of Ist1 orthologs.** (A) GlIst1 of *Giardia lamblia*. (B) Ist1 of S. cerevisiae. The helices are colour-coded to indicate their position relative to the N-terminus: α-1 (red), α-2 (green), α-3 (blue), α-4 (orange), α-4a (sand), α-4b (chartreuse green) and α-5 (magenta) and the MIM (yellow). The unstructured regions are in cyan.
(TIF)

**S7 Fig. Detection of myristoylation in GlIst1 using LC-ESI-MS/MS.** (A) Peptide fragments of GlIst1 showing myristoylation (red) and ubiquitination (blue). (B) Overall detection of peptide fragments is highlighted in yellow, with the myristoylated K residue highlighted in red. The peptide sequence coverage for GlIst1 was 55.8%.
(TIF)

**S8 Fig. A schematic cartoon illustrating the subcellular distribution of Vps4/Vps46 paralogs in trophozoites and 48 h of encysting trophozoites.**
(TIF)

## Acknowledgments

The authors acknowledge all the members of the Sarkar laboratory for discussions and valuable comments. DNA sequencing and confocal imaging were carried out at the CIF of Bose Institute. Prantik Saha and Sheolee Ghosh Chakraborty are acknowledged for assisting with confocal imaging.

## Author contributions

**Conceptualization:** Nabanita Patra, Nabanita Saha, Srimonti Sarkar.

**Data curation:** Nabanita Patra, Nabanita Saha, Trisha Ghosh, Babai Hazra, Shirsha Samanta, Pritha Mandal.

**Formal analysis:** Nabanita Patra, Trisha Ghosh.

**Funding acquisition:** Srimonti Sarkar.

**Investigation:** Srimonti Sarkar.

**Methodology:** Nabanita Patra, Nabanita Saha, Trisha Ghosh, Babai Hazra, Shirsha Samanta, Pritha Mandal, Avishikta Chatterjee, Abhrajyoti Ghosh, Sandipan Ganguly.

**Project administration:** Srimonti Sarkar.

**Resources:** Sandipan Ganguly, Srimonti Sarkar.

**Supervision:** Abhrajyoti Ghosh, Srimonti Sarkar.

**Validation:** Srimonti Sarkar.

**Visualization:** Nabanita Patra, Srimonti Sarkar.

**Writing – original draft:** Nabanita Patra.

**Writing – review & editing:** Srimonti Sarkar.

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
