## [Decision Letter · Decision Letter 0]

21 Apr 2025

Divergent Functions of Late ESCRT Components in Giardia lamblia: Insights from Subcellular Distributions and Protein Interactions

Dear Dr. Sarkar,

Thank you for submitting your manuscript to PLOS Neglected Tropical Diseases. After careful consideration, we feel that it has merit but does not fully meet PLOS Neglected Tropical Diseases's publication criteria as it currently stands. Therefore, we invite you to submit a revised version of the manuscript that addresses the points raised during the review process.

Please submit your revised manuscript within 60 days Jun 20 2025 11:59PM. If you will need more time than this to complete your revisions, please reply to this message or contact the journal office at plosntds@plos.org. Please include the following items when submitting your revised manuscript:

We look forward to receiving your revised manuscript.

Kind regards,

Sudip K Ghosh

Guest Editor

Abhay Satoskar

Section Editor

Shaden Kamhawi

co-Editor-in-Chief

Paul Brindley

co-Editor-in-Chief

**Additional Editor Comments:**

Thank you again for submitting your manuscript " Divergent Functions of Late ESCRT Components in Giardia lamblia: Insights from Subcellular Distributions and Protein Interactions" for publication in PLOS NTD. The reports from the expert referees are attached below. Apologies for the extended time it took, but it was hard to find reviewers who agreed to look at the paper.

As per the reviewer’s comments, they suggest a number of clarifications and new experiments. You can respond to all of the referees’ points - by making the suggested changes or by providing a convincing argument as to why a change cannot or should not be made - then, I encourage you to submit a revised manuscript. In light of the substantial changes requested by these referees, I am considering this as a Major Revision with the understanding that the revised manuscript will be subject to a full review by these same referees. Please note that acceptance of your revised manuscript is not guaranteed.

**Journal Requirements:**

1) We noticed that you used the phrase 'data not shown' in the manuscript. We do not allow these references, as the PLOS data access policy requires that all data be either published with the manuscript or made available in a publicly accessible database. Please amend the supplementary material to include the referenced data or remove the references.

2) Some material included in your submission may be copyrighted. According to PLOSu2019s copyright policy, authors who use figures or other material (e.g., graphics, clipart, maps) from another author or copyright holder must demonstrate or obtain permission to publish this material under the Creative Commons Attribution 4.0 International (CC BY 4.0) License used by PLOS journals. Please closely review the details of PLOSu2019s copyright requirements here: PLOS Licenses and Copyright. If you need to request permissions from a copyright holder, you may use PLOS's Copyright Content Permission form.

Potential Copyright Issues:

i) Please confirm (a) that you are the photographer of Figure S3, or (b) provide written permission from the photographer to publish the photo(s) under our CC BY 4.0 license.

3) Please ensure that the funders and grant numbers match between the Financial Disclosure field and the Funding Information tab in your submission form. Note that the funders must be provided in the same order in both places as well.

**Reviewers' Comments:**

Reviewer's Responses to Questions

**Key Review Criteria Required for Acceptance?**

**Methods:**

-Are the objectives of the study clearly articulated with a clear testable hypothesis stated?

-Is the study design appropriate to address the stated objectives?

-Is the population clearly described and appropriate for the hypothesis being tested?

-Is the sample size sufficient to ensure adequate power to address the hypothesis being tested?

-Were correct statistical analysis used to support conclusions?

-Are there concerns about ethical or regulatory requirements being met?

Reviewer #1: -Are the objectives of the study clearly articulated with a clear testable hypothesis stated?

Yes

-Is the study design appropriate to address the stated objectives?

Yes

-Is the population clearly described and appropriate for the hypothesis being tested?

Yes

-Is the sample size sufficient to ensure adequate power to address the hypothesis being tested?

Yes

-Were correct statistical analysis used to support conclusions?

Yes

-Are there concerns about ethical or regulatory requirements being met?

No

Reviewer #2: No

No

NA

NA

Yes

No

Reviewer #3: No concerns

**Results**

-Does the analysis presented match the analysis plan?

-Are the results clearly and completely presented?

-Are the figures (Tables, Images) of sufficient quality for clarity?

Reviewer #1: -Does the analysis presented match the analysis plan?

Yes

-Are the results clearly and completely presented?

Yes

-Are the figures (Tables, Images) of sufficient quality for clarity?

I believe that the quality of the figures and legends can be improved to make them easier for the reader to understand. For example, it would be positive to increase the cellular structures where each of the molecules was found. I also suggest a scheme that explains in cartoon form the participation of each of the ESCRT paralogs studied here.

Reviewer #2: No

Most except few

Yes

Reviewer #3: No concerns

**Conclusions**

-Are the conclusions supported by the data presented?

-Are the limitations of analysis clearly described?

-Do the authors discuss how these data can be helpful to advance our understanding of the topic under study?

-Is public health relevance addressed?

Reviewer #1: -Are the conclusions supported by the data presented?

Yes. It is a well written article, with enough details to understand the role of ESCRT paralogs in different functions. Furthermore, it presents novel data that will be of interest to the community that studies parasitic protozoans, with emphasis on the ESCRT machinery and its very important roles in various cellular functions.

-Are the limitations of analysis clearly described?

No.

Although the two-hybrid experiments are convincing, I consider that double-labeling experiments to observe and study the interaction of Vps4 and Vps46 paralogs would provide more evidence about their different locations in the cell, which would allow us to hypothesize about their function. Furthermore, western blot assays would reinforce the identity of the proteins

-Do the authors discuss how these data can be helpful to advance our understanding of the topic under study?

Yes

-Is public health relevance addressed?

Yes

Reviewer #2: Mostly except few

No

Yes

NA

Reviewer #3: The conclusions were not supprted by the data presented. Their descriptions on results and discussions (including conclusions) sections seem highly speculated.

**Editorial and Data Presentation Modifications?**

Reviewer #1: It is a manuscript with enough quality to be published in PLOS Neglected Tropical Disease

Reviewer #2: Giardia lamblia, a human gut pathogen, has a minimal ESCRT machinery but multiple paralogs of some late-ESCRT components. Patra et.al. has examined the sub-cellular distribution of Vps4 and Vps46 paralogs, revealing their association with cellular membranes and microtubule structures, suggesting distinct functions. Their redistribution during encystation indicates a role in morphological and functional transitions. In addition, they also characterized GlIst1, an ESCRT-III accessory protein, which is stated to undergo myristoylation, thereby aiding its membrane recruitment. GlIst1 selectively interacts with GlVps4b and GlVps46b, highlighting the unique roles of these paralogs. These findings may contribute towards our basic understanding of the ESCRT machinery in Giardia lamblia. However, concern lies in the lack of any organelle specific controls corresponding to the localization of the stated proteins. Besides several results have been over exemplified without more experimental data corroborating the same. Some specific comments are being provided.

Reviewer #3: (No Response)

**Summary and General Comments**

Reviewer #1: (No Response)

Reviewer #2: Patra et.al, have shown that Giardia lamblia possesses a minimal ESCRT machinery with multiple paralogs of late-ESCRT components, showing distinct sub-cellular distributions and roles. GlIst1, an ESCRT-III accessory protein, is stated to undergo myristoylation and selectively interact with specific paralogs, highlighting their unique functions. Overall the manuscript is well written but the experimental data is inadequate in supporting the claims in the manuscript. Addressing these issues would improve the manuscript considerably.

Major Comments

1. A rationale should be provided as to why encysting trophozoites are observed specifically at 16h and 48h post-induction.

2. The authors analyze the subcellular distribution of GlVps4 paralogs in trophozoites and after encystation induction. At 16 hours, GlVps4a was detected in the PVs, axonemes of the anterior flagella (AF), and other locations. How have the authors determined such specialized intracellular structures such as PV, AF,VD and C inside the parasite without any counterstaining with such structure specific markers?

3. Line 330: The authors claim that "all paralogs of GlVps4 localize to microtubule-rich structures." However, microtubules were not stained. To validate this statement, a co-localization study should be performed.

4. Line 346: If the data supporting this statement is not shown, then it should be removed.

5. Lines 367-368: To substantiate the claim of increased GlVps46a signal, qPCR should be performed to analyze its expression pattern.

6. To confirm the interaction of GIIST1 with GIVps4b or GIVps4b6b, a co-immunoprecipitation (Co-IP) assay should be carried out in homologous system.

7. Line 564-571: the relevant figure or supplementary figure number should be cited. If Figure 6B is being mentioned, how did the authors confirm that GlIS1 localizes to both the endosome and vacuole membranes? The authors should carry out immunostaining against an endosome/vacuole-specific marker to substantiate their claims.

8. The authors examined the GlIst1-GlVps4b interaction in a heterologous system. Yeast two-hybrid is known to be sticky in nature and thereby generate lot of protein-protein interaction artifacts. It would be better if these claimed interactions are validated in homologous system.

9. The authors, using LC-ESI-MS/MS analysis, observe that the K43 residue undergoes a unique myristoylation. It is not described how out of different possibilities they arrive at myristoylation. Additionally it would be helpful if they also use other biochemical methods, such as radiolabeled myristate or a click chemistry-based technique etc to substantiate their claim.

10. Contradictory Statements in Lines 349-351: The claim that there are "MORE cytoplasmic puncta" yet the corresponding figure shows that the "cytoplasmic localization is diminished". This appears to be contradictory hence clarification is needed.

Minor Comments

1. Scale bars are absent in Figures 1A, 1B, 2A, 2B, and 6B.

2. The unit "hour (h)" is missing after "48" in line 288. Please correct this.

3. ‘trophozoite’ panel should be added to Figure 1A for comprehensive visualization.

4. Line 300: The statement, "ESCRT machinery is known to induce membrane curvature…" lacks a citation. A relevant reference should be added to support this claim.

5. Lines 346-348: The source of this statement should be explicitly stated or referenced.

6. The Y-axis labels in Figures 4A and 4D is missing.

7. In lines 552-553, the relevant figure or supplementary figure number should be cited for the LC-ESI-MS/MS data.

Reviewer #3: Major comment:

On the first half of the paper, authors successfully identified the localization of GIVps4 and 46 paralog proteins (Fig. 1&2). These findings give new knowledge in the field. On the other hand, the function of these proteins was not assessed in their experiments. Nevertheless, authors emphasized functional importance of these proteins over the paper, which seems very speculative, not logical interpretation of their results. In the latter half, authors tried to address the interaction between GlIst1 and GlVps4/46 paralogs (Fig. 3-5). They successfully identified that only specific paralogs (GIVps4b and GIVps46b) can interact with GlIst1, in which they also identified the binding sites of the activity. These data beautifully identified the interaction between them. Finally, they found post-translational modification, myristoylation of Ist1 (Fig. 6). However, their impact on Giardia lifecycles was not presented, as well.

Overall, authors identified some important findings about GIVps4 (or 46) in Giardia, however, further experiments, such as blockade, knockdown, or overexpression, are required for their conclusions and discussions. The reviewer felt authors should add some data for assessing functions of these proteins, or restructure overall descriptions in the manuscript as less speculative style before considering the formal assessment for acceptance.

Other comments:

1. In figure 1&2 and related descriptions, localization of GIVps4/46 were explained by the positional relationship to the organelles, such as PV, FL, PF, CF, and AF, however, these organelles were presented by the morphology on DIC, not by the immunofluorescence staining. Also, image resolution seems not enough to tell the exact localization of GIVps4/46 in the organelle.

2. In figure 3 and related description (line 411-415), GlIst1 orthologs were found in some isolates, but not found in the other sequenced Giardia genomes. Also, later experiments were performed using WB strain. How do you think about the impact of the presence/absence of GlIst1 on Giardia lifecycles? Is it still essential for all Giardia, or important only for some Giardia strains?

3. In figure 3B, did authors confirm glist1 gene expression only in “WB isolate”? Please specify the strain used in the RT-PCR experiment.

4. In figure 6 and related descriptions, they tried to address the effects of myristoylation on the cellular distribution of Ist1. GlIst1 was localized to both the endosome and the vacuole membrane, whereas ScIst1 was located within the endosome. Can you say distribution of GlIst1 was affected by myristoylation? The reviewer could not realize why myristoylation has the impact, not the other differences between Gl and Sc from the presented data.

5. “Author summary” does not describe the importance of the study findings on Giardia lifecycle (stage-conversion), but just adds general knowledge about Giardiasis. Authors should explain how this work has an impact on the Giardia lifecycles.

PLOS authors have the option to publish the peer review history of their article (what does this mean? ). If published, this will include your full peer review and any attached files.

**Do you want your identity to be public for this peer review?** For information about this choice, including consent withdrawal, please see our Privacy Policy .

Reviewer #1: **Yes: ** Esther Orozco

Reviewer #2: No

Reviewer #3: **Yes: ** Koji Watanabe

**Figure resubmission:**

**Reproducibility:**



---

## [Decision Letter · Decision Letter 1]

23 Sep 2025

Divergent Functions of Late ESCRT Components in Giardia lamblia: Insights from Subcellular Distributions and Protein Interactions

Dear Dr. Sarkar,

Thank you for submitting your manuscript to PLOS Neglected Tropical Diseases. After careful consideration, we feel that it has merit but does not fully meet PLOS Neglected Tropical Diseases's publication criteria as it currently stands. Therefore, we invite you to submit a revised version of the manuscript that addresses the points raised during the review process.

Please submit your revised manuscript within 60 days Oct 23 2025 11:59PM. If you will need more time than this to complete your revisions, please reply to this message or contact the journal office at plosntds@plos.org. Please include the following items when submitting your revised manuscript:

We look forward to receiving your revised manuscript.

Kind regards,

Sudip K Ghosh

Guest Editor

Abhay Satoskar

Section Editor

Shaden Kamhawi

co-Editor-in-Chief

Paul Brindley

co-Editor-in-Chief

**Reviewers' Comments:**

Reviewer's Responses to Questions

**Key Review Criteria Required for Acceptance?**

**Methods:**

-Are the objectives of the study clearly articulated with a clear testable hypothesis stated?

-Is the study design appropriate to address the stated objectives?

-Is the population clearly described and appropriate for the hypothesis being tested?

-Is the sample size sufficient to ensure adequate power to address the hypothesis being tested?

-Were correct statistical analysis used to support conclusions?

-Are there concerns about ethical or regulatory requirements being met?

Reviewer #2: majorly

Reviewer #3: Well described.

**Results:**

-Does the analysis presented match the analysis plan?

-Are the results clearly and completely presented?

-Are the figures (Tables, Images) of sufficient quality for clarity?

Reviewer #2: to some extent

Reviewer #3: Improved properly according to the reviewers' recommendations.

**Conclusions:**

-Are the conclusions supported by the data presented?

-Are the limitations of analysis clearly described?

-Do the authors discuss how these data can be helpful to advance our understanding of the topic under study?

-Is public health relevance addressed?

Reviewer #2: some times overstretched

Reviewer #3: Improved properly according to the reviewers' recommendations.

**Editorial and Data Presentation Modifications?**

Reviewer #2: Still some revision work is required before final acceptance

Reviewer #3: Accept

**Summary and General Comments:**

Reviewer #2: the supplementary figures have now greatly improved the manuscript. Still some issues are pending which needs to be resolved.

1. Line 452-453: ‘However, an MIT-like domain must be present in GlVps4b as

453 it can functionally substitute for ScVps4’. How can this be true if in yeast two hybrid screening they could not substitute each other. But if it they functionally complement as per Ref 25 cited then why do they not complement each other in the yeast two hybrid assay.

2. It is unclear how the construct GlVps4b1-191 or GIVps46b were expressed inside E.coli for co-purification studies. Same for GlVps4b1-109, GlVps4b30-109 for b-galactosidase assays.

3. It is strange to find that GlVps4b has an non-canonical MIT domain but its interacting partner GIIst has a canonical MIM domain predicted through AlphaFold and the same interacting with non-canonical MIT of GIVps4b. Besides if GIIst has canonical MIM domain identified by AlphaFold then why doesn’t it interact with canonical MIT domain in ScVps4.

4. What is meant by an N-terminal Ist1 domain in a GIst1 protein. Is this domain at the N-terminus also non-canonical to similar proteins of other species.

5. Since GIst1 interacts with GIVps4b via the C-terminus and with GIVps46b with the N-terminus it would be interesting to find if they can be co-purified together as a complex which then would indicate that GIst1 acts as the seeding protein glueing the GIVps proteins together at the membrane.

6. Since GIst1 and ScIst1 do not complement each other the homologous localization of GIst1 in Giardia is more important over its heterologous expression and co-localization with Scist1 in yeast.

7. Were the proteomic profiling of GIst1 carried out using mass spectrometric analysis or was it predicted through bioinformatic tools. It would be good to validate it through mass spectrometric analysis.

8. Could the myristoylation of K33 of GIst1 be important for its interaction with GIVps46b.

Reviewer #3: The paper was scientifically improved according to the suggestions.

PLOS authors have the option to publish the peer review history of their article (what does this mean? ). If published, this will include your full peer review and any attached files.

**Do you want your identity to be public for this peer review?** For information about this choice, including consent withdrawal, please see our Privacy Policy .

Reviewer #2: No

Reviewer #3: **Yes: ** Koji Watanabe

**Figure resubmission:**
---

## [Editor Report · Decision Letter 2]

30 Oct 2025

Dear Prof Sarkar,

We are pleased to inform you that your manuscript 'Divergent Functions of Late ESCRT Components in Giardia lamblia: Insights from Subcellular Distributions and Protein Interactions' has been provisionally accepted for publication in PLOS Neglected Tropical Diseases.

Best regards,

Sudip K Ghosh

Guest Editor

Abhay Satoskar

Section Editor

Shaden Kamhawi

co-Editor-in-Chief

Paul Brindley

co-Editor-in-Chief

I am glad to inform you that your paper has been accepted. I have personally gone through the response to the reviewers comment and satisfied. I think there is no need to send to the reviewer again.

---

## [Editor Report · Acceptance letter]

Dear Prof Sarkar,

We are delighted to inform you that your manuscript, " 

Divergent Functions of Late ESCRT Components in Giardia lamblia: Insights from Subcellular Distributions and Protein Interactions," has been formally accepted for publication in PLOS Neglected Tropical Diseases.

Best regards,

Shaden Kamhawi

co-Editor-in-Chief

Paul Brindley

co-Editor-in-Chief
